# In Vitro Models of Ovarian Cancer: Bridging the Gap between Pathophysiology and Mechanistic Models

**DOI:** 10.3390/biom13010103

**Published:** 2023-01-04

**Authors:** Elliot Lopez, Sahil Kamboj, Changchong Chen, Zixu Wang, Sabrina Kellouche, Johanne Leroy-Dudal, Franck Carreiras, Ambroise Lambert, Carole Aimé

**Affiliations:** 1PASTEUR, Département de Chimie, École Normale Supérieure, PSL University, Sorbonne Université, CNRS, 75005 Paris, France; 2Equipe de Recherche sur les Relations Matrice Extracellulaire-Cellules, ERRMECe, EA1391, Groupe Matrice Extracellulaire et Physiopathologie (MECuP), Institut des Matériaux, I-MAT (FD4122), CY Cergy Paris Université, CEDEX, 95031 Neuville sur Oise, France

**Keywords:** ovarian cancer, epithelial-to-mesenchymal transition, ascites, biological engineering, mechanotransduction, extracellular matrix, shear stress, mechanotransduction, microfluidics, in vitro models

## Abstract

Ovarian cancer (OC) is a disease of major concern with a survival rate of about 40% at five years. This is attributed to the lack of visible and reliable symptoms during the onset of the disease, which leads over 80% of patients to be diagnosed at advanced stages. This implies that metastatic activity has advanced to the peritoneal cavity. It is associated with both genetic and phenotypic heterogeneity, which considerably increase the risks of relapse and reduce the survival rate. To understand ovarian cancer pathophysiology and strengthen the ability for drug screening, further development of relevant in vitro models that recapitulate the complexity of OC microenvironment and dynamics of OC cell population is required. In this line, the recent advances of tridimensional (3D) cell culture and microfluidics have allowed the development of highly innovative models that could bridge the gap between pathophysiology and mechanistic models for clinical research. This review first describes the pathophysiology of OC before detailing the engineering strategies developed to recapitulate those main biological features.

## 1. Introduction

Ovarian cancer (OC) is associated with difficult detection, poor prognosis, and high probability of relapse after surgery or chemotherapy, due to late-stage diagnosis [1,2,3]. This raises several challenges: (i) understanding the biological origin of the disease, (ii) identifying the relevant markers of tumor development, (iii) mapping the phenotypic heterogeneities within primary tumor and metastases, and (iv) controlling the interplay between cells and their 3D microenvironment. Developing biologically relevant in vitro models ahead of preclinical models is a milestone to develop new strategies to tackle these clinical challenges [4,5]. Promising studies have started to emerge and OCs have greatly benefited from recent advances in various fields such as single-cell analysis, 3D culture, and microfluidics. Those innovative models must integrate multiple parameters to recapitulate the progression of the pathological context, which include the multiple cellular populations as well as both the mechanical and biochemical cues. In this review, we detail these parameters by associating the advanced technologies developed for their reconstruction in vitro.

First, we describe the pathophysiology of OCs with emphasis on the heterogeneities from genetic to phenotypic scales. Then, we focus on the interactions of cells with their microenvironment, including the scaffold and the circulating microenvironment. The latter plays a fundamental role in OC because of the build-up of fluid in the peritoneal cavity, the ascites, which contains a variety of cellular and acellular components. This fluid, known to contribute to patient morbidity and mortality by facilitating metastasis and contributing to chemoresistance, has become a hallmark of OC [6]. It is thus of particular importance to integrate fluids in in vitro models, both in terms of composition and shear stress. After a survey of 3D cultures and spheroid models, we focus on the integration of mechanotransduction in in vitro models as generated by the shear stress from surrounding ascites. In particular, we discuss the ability of microfluidics to model the peritoneal cavity with the associated mechanical constraints, enabling control of the shear stress to quantitatively monitor the cell genetic and phenotypic response. The ability to reproduce OC plasticity and dissemination by modulating both biochemical and mechanical cues makes these advanced in vitro models highly promising tools that open new perspectives in biomedical and clinical research.

## 2. The Pathophysiology of Ovarian Cancers

### 2.1. The OC Environment

#### 2.1.1. The Peritoneal Cavity and the Accumulation of Ascites

Epithelial ovarian cancer (EOC) represents more than 90% of diagnosed OC cases, classified into five subtypes: high-grade and low-grade serous carcinoma (HGSOC and LGSOC, respectively), mucinous carcinoma, endometrioid carcinoma, and clear cell carcinoma [1,7]. Because HGSOC is the most predominant form of EOC, it is the focus of this review. Although the biological origin of the disease is not clear, it is suspected to originate from fallopian tumors that rapidly metastasize within the ovaries [8,9,10], or from incorrect wound repair after ovulation and inflammation [11]. The ovaries are covered by the germinal epithelium of Waldeyer, which is composed of one layer of squamous-to-cuboidal epithelial cells. This epithelium delimitates the borders of the ovaries, within the peritoneal cavity, from the same embryonic origin as the peritoneal mesothelium delimitating the coelomic cavity (Figure 1).

The peritoneal cavity contains a thin film of peritoneal fluid composed of water, electrolytes, and other substances derived from the interstitial fluid of adjacent tissues that ensures the harmless movements of the surrounding organs. In a pathological context, an inflammatory liquid—called ascites—accumulates in the cavity due to decreased lymphatic drainage, lymphatic obstruction, and increased vascular permeability [12]. The early presence of malignant ascites has been shown to generally match with poor prognosis, high chemoresistance, and aggressive metastasis [13,14]. It is composed of cancer and non-cancer cells, cell-free DNA, and numerous signaling molecules, as well as extracellular matrix (ECM) proteins and proteases. Additionally, exosomes play a key role in promoting the metastatic niche in this environment [15].

#### 2.1.2. The Cellular Environment

Ovarian primary tumor tissue gathers diverse types of cells, where only a subgroup is defined as malignant. EOC cells are first players, together with other cell types including cancer-associated fibroblasts (CAFs). Cancer cells produce TGF-β, which stimulates interactions with CAFs. CAFs are stromal cells, which contribute to ECM synthesis and remodeling from physiological to pathological environments. This ability of CAFs enables a pro-tumoral activity favoring the metastatic niche [16]. The combination of CAFs ECM remodeling and pro-inflammatory molecule secretion (cytokines, chemokines, growth factors) is responsible for diverse effects from matrix mechanical properties on growth, invasion, and angiogenesis during tumorigenesis [17,18].

Immune cells also play major roles in the tumor microenvironment. In particular, T-cells and tumor-associated macrophages (TAMs) are crucial actors in tumor development. They play a strong immuno-modulating role by producing a large number of cytokines, chemokines, and growth factors. They can hijack the immune response to preserve the tumor [19,20], promote tumor development [21], and even increase metastasis [22,23]. In this review, we do not describe in detail the role of immune cells and rather focus on the impact of cytokine secretions in the surrounding fluid in ovarian cancer. Finally, adipocytes have also been shown to play a role in the tumor environment, acting in metabolite transfer and supporting tumor growth [24,25].

### 2.2. Heterogeneities in OC Cells: From Genetics to Signaling Pathways

In about 10% of cases, OC has a genetic cause. Intensive work has been carried out over the past ten years to screen for genetic mismatches in cancer, including in OC [26,27,28]. One of the extensively studied tumor suppressor genes—TP53—is known to be associated with HGSOC in 96% of cases [26], and mutations in BRCA1/2 increase the risk of developing ovarian cancer by 10 to 25 times. BRCA1/2 play multiple and unique roles in homologous recombination repair. More than 200 different mutations of the BRCA1 gene and 80 of the BRCA2 gene have been listed, 80% of which lead to a non-functional, absent, or truncated protein. Other prevalent genetic mutations implicated in HGSOCs include BRCA1 (12%), BRCA2 (11%), KRAS (6%), NF1 (4%), CDK12 (3%), PIK3CA (2%), and BRAF (2%) [29]. Although some of these mutations are rare, they act as important drivers of HGSOCs. Beyond their role in disease initiation or progression, more than 25 oncogenes have been detected to promote drug resistance [30]. It is also important to mention the impact of epigenetics in the initiation of OC [31].

Most of these mutations directly affect the signaling pathways by modulating the expression of cytokines, including growth factors, and of their receptors through which the signal is transduced [32,33]. Among these, TGF-β is involved in cell proliferation, apoptosis, adhesion, invasion, and angiogenesis. As such, the dysregulation of this pathway results in OC plasticity and leads to OC progression [34,35].

### 2.3. Phenotypic Heterogeneities and the Epithelial-to-Mesenchymal Transition

#### 2.3.1. The Epithelial-to-Mesenchymal Transition

The epithelial-to-mesenchymal transition (EMT) is a morphogenetic process of cellular plasticity that epithelial cells undergo during embryonic development and during several pathophysiological processes such as wound healing, fibrosis, and cancer. During this transition, epithelial cells lose some of their characteristics while gaining mesenchymal features. Among many, epithelial cells present altered cell–matrix interactions due to an associated switch in the population of integrins at the surface of transitioning cells. Similarly, cadherins are also affected on the cell surface, with the up- and downregulation of the N- and E-cadherin expressions, respectively [36,37,38], and the reduction in Ep-CAM expression. Additionally, E-cadherin downregulation combined with occludin can lead to the weakening of tight junctions. This has been proposed to end with the detachment of cells from the ovarian epithelium, involving a role in the dissemination of multicellular aggregates [39,40]. During EMT, the internal cell structure is also remodeled [41,42]. This is illustrated by the change in the expression of α-smooth muscle actin and the nuclear localization of β-catenin. Another illustration of the thorough reorganization of the cytoskeleton is the upregulation of vimentin, switching from a low and perinuclear distribution to a high cortical distribution [43,44]. This transitioning process is not binary, but rather includes a continuum associated with great phenotypic evolutions, well-illustrated by the term “Epithelial-Mesenchymal Plasticity” (EMP) [45]. This makes cells capable of escaping anoikis, a mechanism of cellular death induced by the loss of anchorage.

Ovarian cells are very EMT-active and display an important cellular heterogeneity, as observed within the Waldeyer epithelium after ovulation [46,47]. Indeed, cells from the outer layers of the ovary dedifferentiate and proliferate to create new stroma and epithelium, which fill the damaged area. In a pathological context, this transition has been shown to be partial because of the lack of the classical transcription factors that are typically observed in wounded areas [48]. EOC cells use this innate ability when exposed to promoters such as CAF-secreted TGF-β to detach from the ovaries or fallopian tubes and adopt a floating regime within the ascites (Figure 2) [35,41,49]. Interestingly, the response of cells to TGF-β strongly depends on the cellular subtype and EOC grade [50]. Finally, several articles suggest that this rapid transition can be reversed (the mesenchymal-to-epithelial transition—MET) with cells acquiring epithelial features with strong cell–cell interactions directly on the ovaries [7,51].

#### 2.3.2. The Role of Cancer Stem Cells in EMT

Cancer stem cells (CSCs) are prone to metastatic behaviors and capable of self-renewal [52,53]. It is not known whether this population originates from EOC cells in the course of EMT, or if CSCs appear stochastically with a mesenchymal phenotype to drive tumor growth. Some studies indicate that CSCs possess epithelial characteristics, which supports the first hypothesis [54,55]. Moreover, these cells are shown to have improved capacities for spheroid formation and anoikis resistance [56], but they still lack invasive properties to be considered as the sole actors of the metastatic process [57]. Additionally, their phenotype is different from both primary and metastatic constructs [58]. This suggests that they are the result of several steps of back-and-forth EMT transitions.

The ascitic population is known to house an important proportion of CSCs as compared to tumor tissues [54]. In this pathological fluid, they are identified as key actors, being involved in the survival of floating EOC cells, either as single cells or in aggregates [59,60,61,62]. CSCs act as leader cells that drive the evolution of a whole group, although this population is composed of heterogeneous EOC cells [63].

EMT is thus intrinsically correlated to cellular aggregation, where a balance between retaining epithelial characteristics and developing mesenchymal traits is needed to ensure the survival of a multicellular aggregate and its metastatic outcomes [64]. The non-aggregative population of EOC cells is genetically heterogeneous and cohabits with other cell types within the peritoneal cavity [28]. Because of their ability to undergo EMT and to detach from their anchorage, EOC cells adopt a new phenotype that allows them to survive without tight cell–cell junctions. This switch in their cadherin expression also tends to promote their aggregation, as cells that achieve partial EMT often express a variety of cadherins that enable temporary cell–cell junctions [40,48]. Consequently, these cells are likely to form multicellular spheroid constructs.

#### 2.3.3. Spheroids: Complexity, Invasive Dynamics, and Chemoresistance

Spheroids are formed by self-assembly of cells through different types of intercellular interactions, including cadherin- and integrin-mediated interactions, as well as involving the vitronectin/αv integrin adhesive system. Spheroids contribute to protect cells from the stressful microenvironment, accounting for the resistance to anoikis of cells within spheroids [65]. In vivo tumor spheroids are also characterized by the presence within these multicellular clusters of non-cancerous and non-ovarian cells. Matte et al. described the presence of mesothelial cells in the center of the spheroids (in vivo and in vitro), which makes the spheroids more compact [66]. CAF-tumor spheroids have also been recently reported to promote early peritoneal metastasis of OC [61]. Interestingly, a study from Al Habyan et al. showed that most of the spheroids are likely to come from collective detachment from the ovaries, with little aggregation or proliferation within the ascites [62]. Besides, numerous articles have shown that spheroids are more prone to resist chemotherapies, such as carboplatin and paclitaxel [67,68], with a mechanism that remains unknown. Such organization has hence been suspected of modifying gene expression and increasing aggressiveness [48,69]. However, some argue that spheroids favor the development of CSCs, and that only the latter would be chemoresistant, giving the colony the ability to rapidly regenerate after a conventional drug treatment [69]. Still, spheroids seem to be the preferential mode of survival of floating EOC cells, as well as the starting point for metastatic activity.

### 2.4. The Mechanical Microenvironment of OC: From Scaffold to Circulating Environment

The mechanical microenvironment regulates ovarian cancer cell morphology, migration, and spheroid disaggregation [70]. Indeed, in addition to the interplay between the different signaling pathways, there are interactions with the scaffold and the circulating environments, namely the extracellular matrix (ECM) and ascites. Let us first consider the role of ECM in EOC.

#### 2.4.1. The Extracellular Matrix and Its Mechanosensing

The ECM is a key player in regulating cell migration, differentiation, and proliferation, and is a determinant for the growth and progression of solid tumors. In the ovary, it is made of a variety of molecules including the collagen superfamily, glycoproteins, proteoglycans, and hyaluronan [71]. While the most abundant proteins of the ovarian interstitial matrix include fibrillary collagens (I and III), the basement membrane underlying the ovarian surface epithelium is composed of a dense network of collagen IV and laminin. The ECM is constantly remodeled both in normal and tumor development. Important changes in terms of composition, topology, and stiffness have been reported ex vivo in normal and malignant human ovarian biopsies [72,73]. Basement membrane remodeling in pre-malignant ovarian surface epithelium has been shown to mostly impact collagen IV and laminin with transient loss [74,75,76]. Note that fibronectin and vitronectin also play a key role in ovarian cancer, notably as circulating components of the ascitic malignant fluid.

Apart from its biochemical composition, the ECM is also key in terms of biophysical properties. This is commonly known as mechanosensing. Mechanosensing and the impact of the ECM on cell behavior have motivated an increasing number of works during the past few years [77,78]. In particular, mechanosensing has been shown to influence many cell processes including tumor activity of EOC. For example, Zhou and colleagues have shown that growing SKOV-3 cells in hydrogel of different stiffness modulates the formation of spheroids, the presence of CSC characteristics, and their resistance to doxorubicin [79]. They have highlighted a significant stiffness-dependent increase in the expression of several markers such as ALDH1, which is supposed to be involved both in chemoresistance and in differentiation of EOC cells, as well as of CD117, a receptor tyrosine kinase that allows cells to acquire stem-like properties such as self-renewal, and also CD133, which is known to play a role in EMP.

Furthermore, the invasion of EOC cells has been shown to be dependent on the substrate, with a softer substrate promoting more aggressive and faster invasion according to Dikovsky et al. [80] or McGrail et al. [81], although still controversial [82]. McGrail and colleagues explained that SKOV-3 ovarian cancer cells display a more malignant phenotype on polyacrylamide soft substrates (ca. 3 kPa) undergoing an EMT-characteristic morphological elongation. Single-cell motility analysis revealed large increases in migration on soft substrates, as quantified by the calculated cell migration coefficient (Figure 3A). In addition, they have used traction force microscopy to quantify the force exerted by cells on the underlying substrate. These experiments have shown that when cultured on soft matrices, OC cells exerted more force than when cultured on stiffer polyacrylamide substrates (ca. 35 kPa, Figure 3B), which is attributed to an increased metastatic phenotype. Moreover, the increased intensity and polarization of phosphorylated myosin light chain (pMLC) indicates that cells were more capable of polarizing these forces on the soft substrates (Figure 3C), a crucial step for effective cell migration.

These findings show that the mechanical environment is key to determine cancer progression and highlight how crucial it is to identify key parameters to be reproduced for reconstructing a biologically relevant in vitro environment. This has been highlighted by Pearce and co-workers in their profiling of HGSOC metastases [83]. They integrated gene expression, matrisome proteomics, ECM organization, biomechanical properties, and cytokine and chemokine levels all from the same sample. By doing so, they attempted to predict the extent of disease, while also revealing the dynamic nature of matrisome remodeling during tumor development. Further studies are probably needed to better identify the range of mechanical forces involved and to always increase the level of complexity by integrating the interaction with the topology of the environment, the biomolecular composition, without neglecting the circulating environment.

#### 2.4.2. The Importance of Fluids in OC: Soluble Factors and Shear Stress

Ascites acts as a regulator of cell–ECM interactions, exposing cells to a high concentration of ECM proteins. In particular, integrin α5β1 and αvβ3 ligands are believed to be a preferential path of adhesion for EOC cells to the ascites fibronectin and vitronectin, respectively [84,85]. We and others have previously shown that mesothelial secretions, including vitronectin, promote the invasion of OC cells [85,86,87,88]. However, these findings are still controversial [89], which can partly be attributed to the diversity of cell lines and the heterogeneity in patient ascites. This highlights the fact that further studies have to be conducted on the interactions between invasive cells and proteins in the ascites.

Biophysical inputs are also at play in the circulating environment. Indeed, Bascetin et al. have shown that the macromolecular crowding (MMC) of the ascites microenvironment impacts OC cell phenotype [90]. In their recent work, they used two inert crowders, Ficoll 400 kDa and Dextran 250 kDa, to mimic the MMC and the estimated total protein concentration in OC ascites. They notably looked at the effect of extracellular MMC on actin organization of two different OC cell lines. Without MMC, SKOV3 and IGROV1 cells were well spread, with peripheral cortical actin and/or central stress fibers (Figure 4A). In contrast, in the presence of crowders, cell spreading significantly decreased with much reduced cortical actin and stress fibers. In this line, ascites viscosity was previously shown to be a marker of the degree of cancer malignancy and correlated with metastasis speed and with the concentration of floating cells [91]. This illustrates the importance of ascites MMC and viscosity when designing a relevant in vitro model of the peritoneal cavity. Both can indeed decrease the metastatic potential, as crowding can inhibit spheroid formation and viscosity can reduce the invasion rate due to limited diffusion.

Specifically, ovarian cells are exposed to a continuous fluid shear stress imposed by diaphragm and organ movements, as well as ascites build-up in advanced OC stages. This makes shear stress dependent on the increase in ascites volume in the peritoneal cavity of each individual patient. An interesting study by Klymenko and colleagues evidenced that the increase in ascites volume triggers a dramatic increase in the intraperitoneal pressure that EOC cells are subjected to, which has few effects on their proliferation, but modifies their metastatic abilities, notably their cadherins expression [92]. This also points out the impact of shear stress as a potential inducer of tumor cell cycle modification or arrest. Chang et al. found that shear stress of 12 dyne/cm² led to a G2/M arrest in four lines of cancer cells, as opposed to their control in static conditions [93]. In a systematic study, they characterized the mechanisms by which shear stress regulates cell cycle in tumor cells, proposing specific roles to integrins and Smad proteins (Figure 4B). They hypothesized that the in vivo behavior of these cells could result in the preferential invasion of areas with smaller, laminar shear stress. However, a precise magnitude of shear stress within the peritoneal cavity is still difficult to obtain. Most of the articles now agree on the upper barrier [94,95,96], which could be placed at some units to 10 dyne/cm².

Modeling and integrating key parameters of the cell microenvironment is a crucial step to foster progress in biological and biomedical research, and particularly here for investigating cell EMP, migration, and metastasis progression. In the last part of this review, we will discuss how the different characteristics of the scaffold, which are all known to affect cells, must be controlled and reproduced using biomaterials approach for designing in vitro models. In addition to these crucial aspects, we will also discuss the need for integrating co-culture, 3D models, and the fluid microenvironment.

## 3. Reconstructing OC in Its Multidimensional Environment

New tools are constantly being developed to always improve the biological relevance of in vitro models. This comprises 3D culture setups and co-cultures, which represent a great improvement as compared to traditional 2D culture models. Recent smart culture setups also include hydrogel seeding, multilayered constructs, droplet culture, and spheroid growth. Additionally, microfluidics and additive fabrication have also come across to enrich the battery of possibilities, easing the engineering and vascularization of 3D culture models.

All these tools and strategies should be used in a complementary fashion to reach the highest possible integration level. This is illustrated in Figure 5, which highlights the necessity to increase the integration of the in vitro model, starting from ECM models, 3D cellular models, and recapitulating the circulating environment, in terms of composition, dynamics, and shear stress. The tools developed in the past decade to tackle these issues are discussed in the following sections.

### 3.1. Co-Culture, Which Cells?

Important research efforts have been dedicated to ovarian carcinomas, either to provide a reliable model on which screening assays could be conducted for drug development [97], or to build up an integrative model to mimic and better understand OC physiology [98]. As discussed before, several cell types are involved in OC such as CAFs, mesothelial and EOC cells. EOC cells are first players, widely commercially available for their use in biomedical research, and with significant differences in behavior and expression [46]. Among those, SKOV3 cells have the advantage of exhibiting an innate plasticity, which makes it a relevant model cell line for EOC [47]. One step further, EOC cells’ co-culture is the first challenge to meet for building in vitro models. In this line, several groups have reported the growth of EOC cells onto a layer of mesothelial cells to assess the influence of mesothelial secretions [88,99,100].

Besides mesothelial cells, the influence of adipocytes on EOC metastasis has also been a focus of interest. Indeed, adipocytes have been shown to increase proliferation and chemoresistance of EOC cells, and may constitute a preferential invasion site for floating aggregates [24,81,101,102]. On this basis, multi-cellular culture models were developed by Pearce and Balkwill [103,104,105]. In their study on the impact of platelets on extracellular matrix production and tissue invasion, they used a 3D model where fibroblasts were plated on top of a gel loaded with adipocytes, followed by a layer of mesothelial cells. This multi-cellular construct was cultured for 24 h. Then, a tetra-culture construct was obtained by the addition of HGSOC cells (Figure 6A) [105]. Ultimately, a further level of integration was reached by the addition of fresh isolated platelets. Figure 6B–D reproduces representative confocal images obtained after 7 days. These images show that ECM molecules such as fibronectin (FN1) and versican (VCAN) were present to a higher level on the pentaculture construct. In addition, quantification of EpCAM deposition demonstrates the presence of a higher number of malignant cells in the penta-culture (Figure 6E). This was further confirmed by flow cytometry analysis of EpCAM-positive cells (Figure 6F). This indicates that platelets stimulate the production of ECM molecules and malignant cell invasion, further associated with poor prognosis.

Immune cells should also be included in co-culture setups to investigate the role of tumor-associated macrophages (TAMs) in tumor growth. Indeed, such works have shown that OC cells and macrophages interact via cell–cell contacts within spheroids, contributing to the switch towards malignant EOC phenotypes [21]. More generally, this provides a platform to study the deregulation of the immune balance in a pathological environment [106]. Finally, pioneering works were also reported by Lengyel, Kenny, and co-workers at Chicago University with the design of organotypic models of the peritoneal cavity [27,107,108,109]. These are composed of primary human omental fibroblasts mixed with ECM components (fibronectin and type I collagen) and covered with a confluent layer of mesothelial cells, on top of which EOC cells were seeded. These works have allowed a great advance in the description of the impact, and hence in the design, of the properties of the matrix in a cellular model with increased biological relevance. As such, these works are further discussed in Section 3.3.

Co-culture models are hence of high relevance to better reproduce the pathophysiological context, and to better understand the interactions between several cell types, with an increasing consideration for 3D cellular constructs [110].

### 3.2. Spheroid Models

Many researchers have reported that cell behavior, including OC cells, and response to drugs are different in 2D monolayers and 3D spheroid models [110,111]. This highlights the challenges of choosing the appropriate pre-clinical models for drug testing. As a result, most of the current researches in the field elaborate protocols to grow cells in 3D constructs [112]. These techniques involve the seeding of cells on non-adherent substrates such as bovine serum albumin (BSA) [113,114], polyHEMA [95,115], agarose [65,116,117], or Pluronic F127 [118,119] or the use of EOC cell lines known to spontaneously aggregate in given conditions [46,120]. Alternatively, spheroids can be produced by using hydrogels [79,81,121,122,123,124], hanging drop culture [125,126], or rotating wall culture [94,127,128] that have all proven their efficiency in the past ten years for the generation of reproducible and stable organoids [129,130]. Alternatively, we have recently reported the development of microfabricated supports to engineer human OC spheroids (Figure 7A–C). We showed that playing with the dimension of the support allows tuning the spheroid size in a controlled and reproducible manner (Figure 7D–F) [131].

Single cell approaches have revealed the stemness potential of spheroid cell populations. Those findings are relevant for cancer cell phenotypic heterogeneity, as well as for drug resistance [132]. Alessandri and co-workers have investigated the impact of confinement on the internal cellular organization of spheroids by encapsulating cells in an aqueous core enclosed by a hydrogel shell [133]. DAPI staining shows that the nuclei are smaller in confined spheroids (Figure 8A,D), leading to a cell density twice as large as in the freely growing spheroids (Figure 8C,F). They also examined cell proliferation by staining with KI-67, showing that proliferative phenotypes are homogeneously distributed throughout free spheroids (Figure 8A,C), contrary to confined spheroids, where cell division mostly occurs at the periphery (Figure 8D,F). Finally, fibronectin was found to be fibrillar and homogeneously distributed in free spheroids, whereas it was restricted to the periphery in confined spheroids (Figure 8B,E). One step further, and by performing invasion assays in a collagen matrix, they have reported that peripheral cells readily escape pre-confined spheroids, while cell–cell cohesion is maintained for freely growing spheroids. This suggests that mechanical cues from the surrounding microenvironment may trigger cell invasion from a growing tumor [133].

Finally, Zhou et al. observed a progressive decrease in the growth of their multicellular aggregates when cultured within hydrogel fragments, which can be explained by the limited oxygen and nutrient diffusion in a crowded environment [79]. These results are an additional illustration of the impact of the matrix and importance of cell–matrix interactions in driving cell behaviors, including cell growth and invasion. This then requires to consider the scaffolding environment of cells to be reconstructed.

### 3.3. ECM and Scaffold

Among different features, the biochemical composition, topology, and mechanical properties of the ECM scaffold are crucial and must be scrupulously monitored and reproduced. This requires advanced techniques of biofabrication. Polymer science and biomaterials research have greatly improved our ability to engineer biological matrices. Among these, electrospinning, which is an electrically assisted extrusion method, is a promising cost-effective technique to mimic the fibrillar aspect of the ECM, which has been shown to influence cell behavior [134,135,136]. Moghadas et al. used electrospinning to design layers of ECM fibers that can host cells for 3D culture and proliferation [137]. In another work, a thick electrospun matrix could further be integrated within a microfluidic chip to study cell invasion throughout the ECM [138]. Alternatively, bioprinting methods enable a great spatial control on cells and ECM patterns and allow making complex structures with layers of mixed compounds and/or different cell types [139]. Recent works have also shown that the materials used and the way they are printed has an influence on cell behavior [140]. This confirms that ECM is a key player in the development and migration of normal and tumor cells [121]. This is further amplified by the findings of Zhou et al., who have shown that the molecular nature of the cell–matrix bindings also determines cellular spreading [124].

Cell–matrix interactions involve biomolecular recognition with specific ECM proteins. This has been thoroughly investigated by Kenny and co-workers, who have studied the impact of different ECM proteins on the adhesion and invasion of OC cells [17]. From these assays, they reported that SKOV3 cells preferentially adhered to and invaded collagen I, followed by binding to collagen IV, fibronectin, vitronectin, and laminin 10 and 1 (Figure 9).

Important research efforts have been dedicated to the investigation of the dynamic behavior of EOC cells on biomimetic substrates to understand how tumors spread throughout the peritoneal cavity. Invasion assays have been proposed that consist of a monolayer of mesothelial cells cultured over a decellularized human tissue or a biocompatible substrate (mostly hydrogel) [120,141,142]. This bi-layered barrier is then used to isolate EOC cells from a chamber filled with FBS to create a chemotactic gradient. By doing so, these teams were able to scrutinize the ability of EOC cells to invade the mesothelial layers by weakening and breaking cell–cell bindings [63]. One step further, other works reported that the ascites fosters this invasion process thanks to several proteins [143]. In parallel, advances in hydrogel synthesis have allowed research groups to build 3D substrates, which migrating cells modify with secreted proteins (including matrix metalloproteases and lysyl oxidases) [144]. Such hydrogel systems also provide models to study contraction effects within the 3D network in the course of cell migration [145].

Finally, the impact of the mechanical properties [146] and more recently, the impact of the topology have been thoroughly studied in different cancer types [138,147,148]. In particular, cells have been shown to modulate their morphology and invade the matrix with a speed that depends on its topology: by using the same matrix with two different fiber sizes, Eslami Amirabadi et al. observed that cells formed larger protrusions onto matrices with smaller fibers, a phenomenon attributed to the maximization of the cell–matrix interface [138]. In parallel, Guzman et al. found that the viscoelastic properties of the matrix, as well as its pore size (given that it is not sub-nuclear in order not to impair cell migration), had little influence on the invasive distance that OC cells travel [149]. They propose that the invasion is mostly determined by the polarization of OC cells along the preferential orientation of the fibers in the matrix.

Investigating the impact of the matrix on tumor growth and progression is still an important challenge that should now be considered with the integration of flow. This adds a dynamic control over the circulating environment and reproduces shear stress at play in the peritoneal cavity.

### 3.4. Contribution of Microfluidics: Consideration of Shear Forces in Mechanotransduction

#### 3.4.1. Examples of Microfluidic Setups

Although microfluidics has been more traditionally used as a tool to increase the throughput of anticancer drug assays [118,126], it is now becoming widely used to grow and differentiate spheroids or embryoid bodies [113,150], as well as for building in vitro cancer models with flow control for mimicking body flow and vasculature [151]. In particular, microfluidics has been considered as a candidate to reproduce the complex interplays within the peritoneal cavity during OC. Since first reported chips with a unidirectional perfusion and separated inlet and outlet reservoirs for co-culturing [152], different setups have been reported for investigating OC (Figure 10). Li and co-workers reported the engineering of a 3D ovarian cancer–mesothelium microfluidic platform, where OC spheroids were co-cultured with primary human peritoneal mesothelial cells. In this chip, mesothelial cells were plated on fibronectin. Non-adherent OC spheroids were seeded into microfluidic channels with continuous flow medium perfused by a syringe pump (Figure 10A) [153]. This setup allowed them to study the impact of shear stress on spheroids, as well as the interplays between fibronectin, mesothelial cells, and EOC spheroids, as expected in the peritoneal cavity during metastasis. In another work, Rizvi et al. designed a linear microfluidic chip to study the impact of flow on the attachment and growth of OC cells [154]. In their setup, tumor cells entered the channels through gas permeable silicone tubing and flowed over stromal beds of growth factor reduced (GFR) Matrigel. A portion of the cells are effluxed from the chip via the outlet tubing (Figure 10B). Those that adhered to the Matrigel beds were cultured under continuous flow (shear stress of 0.5 to 3 dyne/cm²) for seven days. In this work, they reported the flow-induced, transcriptionally regulated decrease in E-cadherin protein expression and the simultaneous increase in vimentin, indicating an increasing metastatic potential.

As a last example, Avraham-Chakim et al. developed a flow chamber for direct application of fluid flow induced wall shear stress (WSS) on a monolayer of OVCAR-3 OC cells cultured on denuded amniotic membranes (mainly collagen and fibronectin) [155]. The chip could be disassembled to install the membrane in custom-designed wells and the cells in the testing flow chamber, and then re-assembled for biological testing (Figure 10C). The flow chamber was designed to hold three wells hosting cells for multiple experiments. The pump could generate a steady flow with a uniform field of shear forces on top of the cells (0.5, 1, and 1.5 dyne/cm²). The space under the well-bottoms inside the flow chamber was filled with static culture medium in contact with the bottom plane of the membrane. From this work, they have suggested that WSS has a significant impact on the mechanical regulation of EOC spreading in the peritoneal cavity by acting on cytoskeleton reorganization (cell elongation, stress fibers formation, and microtubules generation).

#### 3.4.2. Contribution of Microfluidics in Understanding the Progression of Ovarian Cancer

While the pattern and magnitude of fluid motion in the peritoneal cavity remains difficult to map precisely, the shear stress on cells is estimated to be in the 0–10 dyne/cm² range [94]. This flow-induced mechanotransduction triggers rapid signaling events, which impact cytoskeleton organization and further drive cell proliferation, adhesion, and invasion.

##### How Microfluidic Shear Stress Impact Ovarian Cancer

In their work dating back to 2010, van der Meer and co-workers have shown how microfluidics can provide a mechanistic insight into cytoskeleton remodeling when cells—endothelial cells in this case—experience directional shear stress [156]. In the context of ovarian cancer, Hyler et al. proposed more recently to use swirling and rotating fluid circulation to better mimic peritoneal fluid motion [94]. In addition, they have exposed cells to repetitive 96 h periods of fluid shear stress on disseminated and adherent OC cells. This setup is supposed to mimic the physiological exfoliation of surface cells in the peritoneal cavity, which could re-adhere at a different location. They used mouse ovarian cancer epithelial cells (MOSE) from benign (MOSE-E), slow (MOSE-L), and fast (MOSE-LTICν)-developing cancer together with human SKOV-3 OC cells. After exposure to fluid shear stress, they observed that actin protrusions were increased in all tumorigenic cells (Figure 11A). In parallel, they investigated the shear-induced changes in cell adhesion by determining the number and length of vinculin-containing focal adhesions. Overall, cells exhibit a higher number of focal adhesions under shear stress, which was particularly significant for the benign MOSE-E cells. However, the focal adhesion length was found to slightly increase in tumor cells experiencing flow, contrary to the benign MOSE-E cells (Figure 11B). This is in line with the work of Avraham-Chakim et al., which has evidenced shear stress-induced cytoskeleton reorganization of OVCAR-3 OC cells cultured in flow chambers coated with denuded amniotic membranes (Figure 10C) [155].

In their work, Hyler et al. also showed that cells exposed to fluid shear stress exhibited a high increase in CREST-positive micronuclei, which signed for chromosome mis-segregation during mitosis. Finally, they have reported that tumorigenic OC cell lines responded to shear stress by detaching and forming spheroids [94]. The formation and stemness of OC spheroid have also been investigated in another microfluidic chip by the group of Wong [96]. Their device provides a continuous well-defined flow rate (0.002 and 0.02 dyne/cm²), with a synthetic polymer coating (poly-HEMA) preventing cell attachment and matrix deposition (Figure 12A). This aims at keeping tumor spheroids in suspension, as observed in patient ascites. From this setup, they have reported that spheroids under flow expressed stem cell markers (Oct-4, c-Kit, ABCG2, and P-gp), contrary to what was observed under static conditions. Furthermore, they detected an enrichment in CD117^+^/CD44^+^ cells in spheroids exposed to shear stress, together with enhanced self-renewal potential, differentiation ability, and increased tumor-initiating capability. Finally, they identified a mechanosensitive miRNA—miR-199a-3p—that showed a marked decrease under physiologic shear stress [96]. These findings stimulated further works on this signaling pathway using the same polymer-coated microfluidic device. Working with a shear stress of 0.02 dyne/cm², they identified c-Met as a shear stress-responsive receptor tyrosine kinase [154]. This provides a mechanistic insight into the downstream regulation of miR-199a-3p and the consecutive impact on drug resistance. These results confirm previous reports from Rizvi et al., who described the increased biomarker expression and tumor morphology consistent with increased EMT, and that is attributed to hydrodynamic forces (Figure 10B) [154]. These findings indicate that fluid shear stress induces a motile and aggressive tumor phenotype, which is driven in part by a post-translational upregulation of epidermal growth factor receptor (EGFR) expression and activation, in turn associated with the worst prognosis in ovarian cancer.

While this illustrates how microfluidics enable addressing the interplay between biophysical and biochemical cues in the ascitic microenvironment, it also highlights the necessity to develop advanced devices for therapeutic development.

##### Shear Stress in Chemoresistance

Wong and his research group have devoted research efforts to understanding the impact of fluid shear stress on chemoresistance in ovarian cancer. Their device provides a continuous well-defined flow rate (0.002 and 0.02 dyne/cm²), with a synthetic polymer coating (poly-HEMA) preventing cell attachment and matrix deposition (Figure 12A) [96]. Spheroids were grown from SKOV3 cells with a mean diameter of 104.6 ± 1.67 µm. These spheroids were treated with two antitumoral drugs cisplatin and paclitaxel, in the presence or absence of shear stress. The results are analyzed with an Annexin V/PI staining to detect viable, necrotic, early and late apoptotic cells (see Figure 12B). In static conditions, cells in OC spheroids rapidly underwent apoptosis upon cisplatin and paclitaxel treatment (upper right panels, Figure 12B). In contrast, under flow, cells in tumor spheroids showed significantly greater chemoresistance in the presence of cisplatin and paclitaxel, with 65% to 70% of cells in spheroids remaining viable (lower left panels, Figure 12B).

Deciphering the mechanisms of chemoresistance remains an open challenge. Still, the PI3K/Akt signaling is of particular relevance for chemoresistance. In recent works, Wong’s group again deepened this understanding of the inverse correlation of miR-199a-3p expression with enhanced drug resistance in chemoresistant OC cell lines [157]. In particular, they proposed that the shear stress-dependent downregulation of miR-199a-3p expression may activate PI3K/Akt signaling. These results should provide an additional key for correlating stemness and chemoresistance in OC, and hopefully represent a viable target for therapeutic development.

#### 3.4.3. Vascularized Microfluidic Models

Microfluidics has been applied to the world of biology and medicine to build models where cells are confronted with the flow conditions of living systems. This concerns the shear rate and the resulting mechanotransduction pathways. It is also about the supply of nutrients and oxygen to cells. In other words, it is about mimicking the vascular system and its essential role in the optimal functioning of organs.

In this context, the group of Jain built an organ-on-a-chip model of ovarian cancer dedicated to the investigation of the cross-talk between vessels, platelets, and ovarian cancer cells. In a first setup, called the ovarian cancer-on-chip (OvCa-Chip), they superimposed two PDMS fluidic chambers [158]. The top microchannel is seeded with human ovarian A2780 tumor cells and mimics the peritoneal cavity. The bottom channel is lined with human primary endothelial cells that form continuous monolayers and cover all four sides of the microchannel, creating a blood-perfused vessel. One step further, they have incorporated a collagen-based ECM adjacent to the tumor cell chamber, ending with a so-called ovarian tumor microenvironment organ-on-chip (OTME-Chip) (Figure 13A) [159]. Freshly derived platelets from human blood were perfused through the bottom microchannel for 3 days and cell invasion dynamics were monitored with respect to platelet extravasation (Figure 13B). Using gene-edited tumors and RNA sequencing, they investigated the impact of the interactions between glycoprotein VI (GPVI) and galectin-3 in mediating platelet-promoted tumor metastasis. To this end, they compared the OTME-Chip seeded with OC cells with a chip involving galectin-3 knocked out ovarian cancer cells (KO-OTME-Chip). Further comparison was performed with a Control-Chip, where no platelet was perfused. Rapid ECM invasion was seen in the OTME-Chip compared with the KO version, indicating a role of GPVI and galectin-3 in the platelet-promoted metastasis. The role of platelets in promoting ovarian cancer metastasis was further confirmed by the even lower invasion observed in the platelet-free Control-Chip (Figure 13C,D). Finally, they perfused the GPVI inhibitor Revacept, which impaired metastatic potential, illustrating the possibility of using the OTME-Chip for therapeutic exploration. From a technological point of view, this work shows how important it is to associate vascularization to validate in vitro models.

From vascularized in vitro chip models, a further step in integrating multicellular constructs has been recently achieved by Ibrahim and co-workers [160]. In their work, they co-cultured tumor cells (TCs) with mesothelial and endothelial cells (MCs and ECs), together with adipocytes (ADs) (Figure 14A,B). The 3D aspect of the model is ensured by the use of a fibrin hydrogel seeded with ADs providing TCs with a rich stroma. This fibrin gel can be vascularized by ECs and topped with a monolayer of MCs (Figure 14C). The combination of the four different types of cell in a 3D model enable investigating the impact of the permeability of the mesothelial monolayer, as well as the impact of vascular permeability in promoting intraperitoneal metastases.

In a first step, they investigated the ability of cells to secrete ECM proteins in the chip-integrated 3D model. To this end, they checked the expression of collagen VI and fibronectin, which are highly expressed in the ECM of the omentum and peritoneum (Figure 14D). Then, TCs were seeded on the mesothelial layer with varying cell densities (Figure 14E). By doing so, they have shown that a critical cell density is required for tumor growth. In addition, tumor growth was further enhanced by stromal ADs and ECs present in the peritoneal omentum. As a result, and beyond vascularization, this work shows that multicellular 3D models are essential to elucidate tumor–stromal cell interactions during intraperitoneal metastasis of ovarian cancer.

## 4. Conclusions and Perspectives

Ovarian cancer as a living system is multifaceted and multiparametric, notably combining biochemical and biophysical cues. It is also the crossroads of many pathways and the seat of a heterogeneity—including cellular—which remains difficult to describe at present. This makes it a huge challenge to tackle. A crucial direction to take to meet this challenge is to work on the design of biologically relevant models. What parameters should in vitro models integrate to be relevant for bringing new findings in a pathophysiological context? It is a whole scientific field that brings together researchers with different expertise, working together for a common objective. This line has seen the development of biological and biomedical engineering to fabricate model systems (3D cell culture models, spheroid engineering, scaffold-induced mechanotransduction) and associated physical and metric instruments to describe cellular behaviors. More recently, microfluidics has been introduced to the world of cell biology through high throughput and drug screening. Moreover, microfluidics has for years brought new perspectives in the study of cancer thanks to the integration of flow, which is now recognized as a crucial actor in the biological context, both in terms of circulating factors and shear-induced mechanotransduction. Research effort on ovarian cancers, among others, has strongly emphasized the need to develop devices based on flow for the development of therapeutics.

Microfluidics enable fine tuning of parameters to model the influence of specific signals from the microenvironment in cellular heterogeneity. We believe that the combination of quantitative tools for monitoring and measuring cellular parameters with microfluidics constitutes the Rosetta stone for deciphering the complex interactions of the microenvironment and the cells involved in the pathophysiological processes of ovarian cancer. A challenge remains in the ability to control and switch those interactions not only externally with microfluidics, but from the cell perspective. Controlling cell interaction with the microenvironment with microfluidic-like speed and precision is a key challenge to achieve this level of decipherment.

Microfluidics has already contributed to the identification of several general mechanisms that govern processes during cancer progression, including tumor growth, metastasis, and chemoresistance. Questions that have been discussed in this review still remain to be mastered. In particular, the possibility of increasing the integration of different cell types, and the proper measurement of each in a so-called co-culture system should always be improved and deepened. Other aspects beyond the scope of this review are worth mentioning here in the perspective of ever better recapitulating living systems. To name a few, one can emphasize the need to reproduce and control the dynamic remodeling of the extracellular matrix during tumor progression. Moreover, the communication with distant organs must also be modeled. This must be based on the possibility of vascularizing the in vitro models. This is another active area of research in microfluidics and organ-on-chip, whose maturity should make it possible, in the near future, to make a leap forward in understanding the mechanisms of tumor progression, in particular by repositioning ovarian cancer in a whole organism.

## Figures and Tables

**Figure 1 biomolecules-13-00103-f001:**
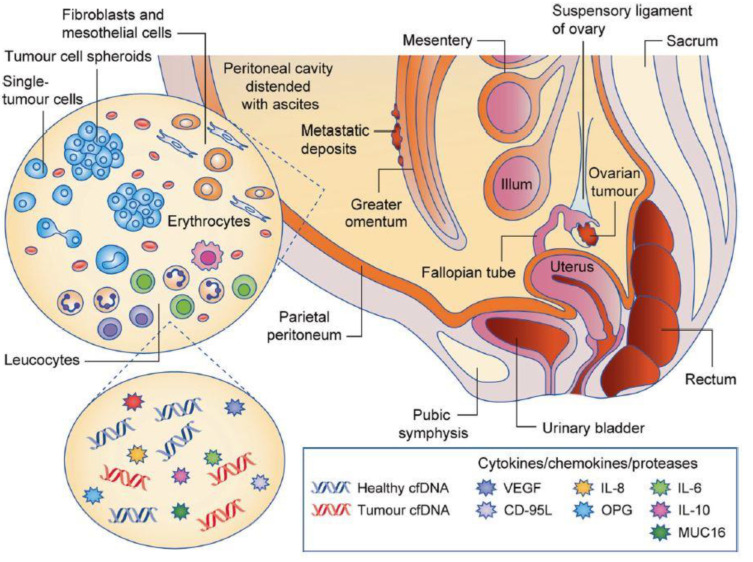
Scheme of the peritoneal cavity, which contains the ovaries and is surrounded by the parietal and visceral leaflets [6]. The abnormal increase in ascites volume lines with enrichment in pro-inflammatory cytokines, circulating nucleic acids, and various cell types, as well as proteins from the extracellular matrix. Reproduced with permission of Springer Nature.

**Figure 2 biomolecules-13-00103-f002:**
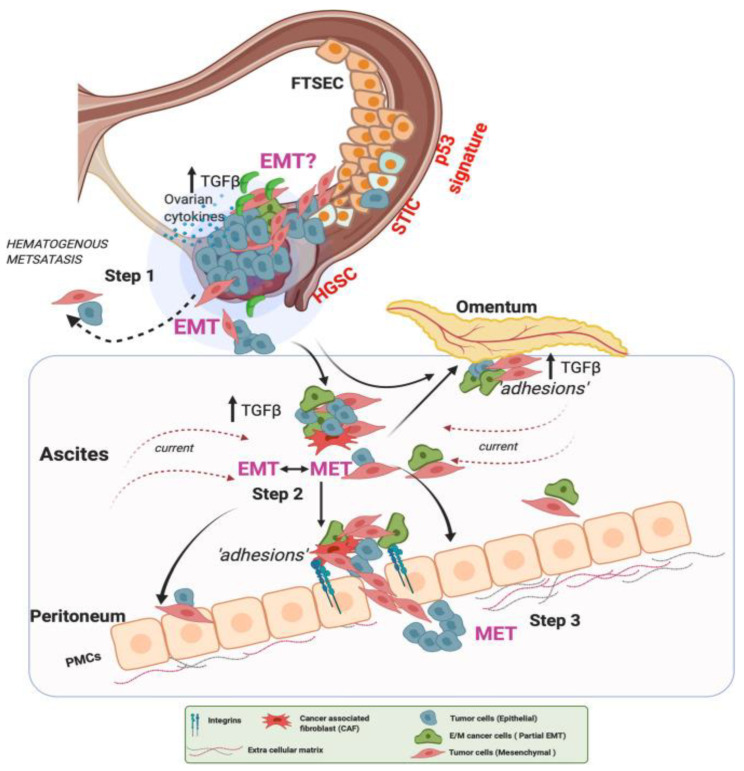
EMT events in ovarian cancer hematogenous metastasis. In Step 1, lesions with characteristic alterations in TP53 develop into cancers in the fallopian tube and the ovaries. OC cells detach and shed into the peritoneal fluid for transcoelomic spread or enter the blood vessels leading to hematogenous metastasis. In Step 2, OC cells in the ascites show high heterogeneity along the EMT continuum, forming anoikis-resistant cell aggregates. Ascites facilitates cell aggregate adhesions to the peritoneal membrane. Such adhesions in Step 3 can undergo MET (reverse EMT), enabling the cells to establish and grow at secondary sites. At the peritoneal interface, cancer cells invade peritoneal mesothelial cells facilitated by integrins and TGF-β, developing secondary tumors and metastasis [35]. Reproduced with permission of Springer Nature.

**Figure 3 biomolecules-13-00103-f003:**
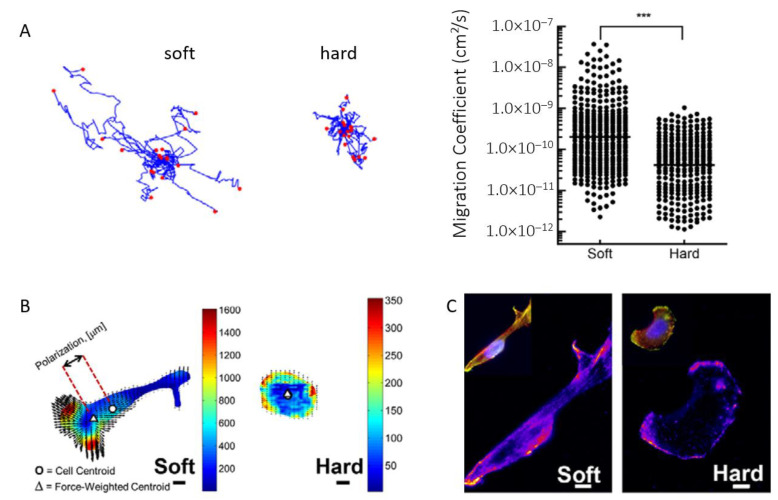
(**A**) Cell motility tracking on soft and hard matrices and analysis revealing a significantly higher cell migration coefficient on soft substrates. (**B**) Heat maps of traction stresses (Pascals) overlaid with black arrows showing cell-induced matrix displacements. The cell center of mass is shown by the circle, and the triangle shows the force-weighted center of mass. Scale bars: 10 µm. (**C**) Staining for pMLC reveals a corresponding increase in pMLC intensity on soft substrates. Data are shown as the mean ± s.e.m.; *** *p* < 0.001. Reproduced with permission from [81].

**Figure 4 biomolecules-13-00103-f004:**
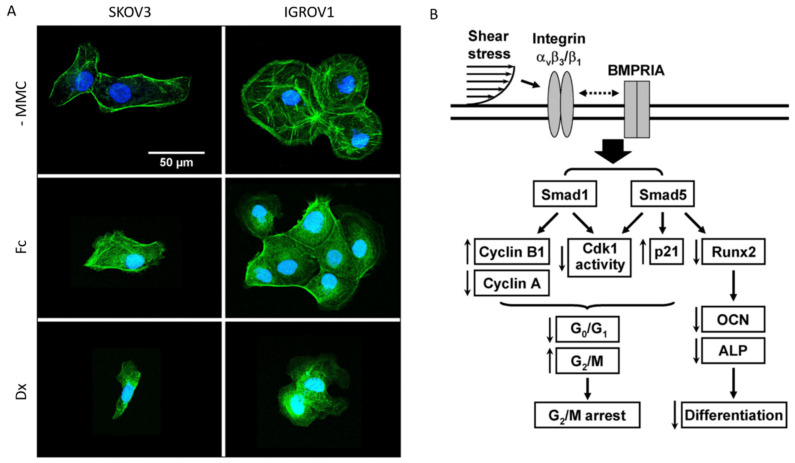
(**A**) Adherent SKOV3 and IGROV1 cells were starved overnight in serum-free medium, and cultured 6 h in their culture media supplemented with 0 (-MMC) or 75 mg/mL of Ficoll 400 kDa (Fc) or Dextran 250 kDa (Dx). Cells were stained for actin (green) and DNA (blue). Representative images of two independent experiments in duplicate reveal the reorganization of the actin cytoskeleton in crowded environments. Reproduced with permission from Elsevier [92]. (**B**) Schematic representation of the signaling pathways regulating cell cycle and differentiation in tumor cells in response to shear stress. ↑ up and ↓ downregulation by shear; dotted double-arrow line represents the interaction pathway that has not been defined. Copyright (2008) National Academy of Sciences, USA [93].

**Figure 5 biomolecules-13-00103-f005:**
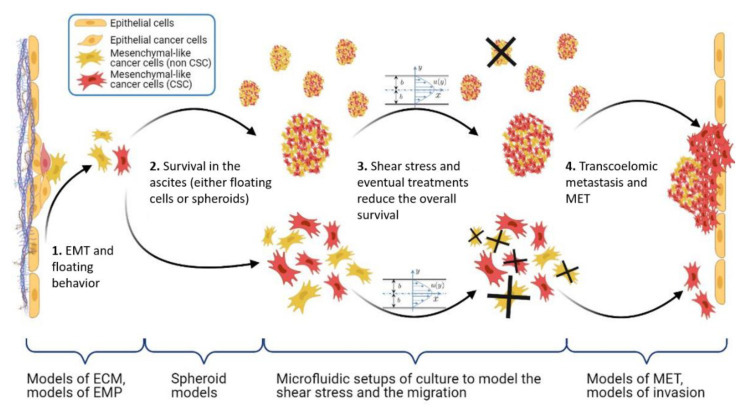
Scheme of the main steps of the OC metastasis process: EMP, stemness, and shear stress. Under braces are the different in vitro models, which aim at recapitulating the corresponding pathophysiological context. Made with BioRender.

**Figure 6 biomolecules-13-00103-f006:**
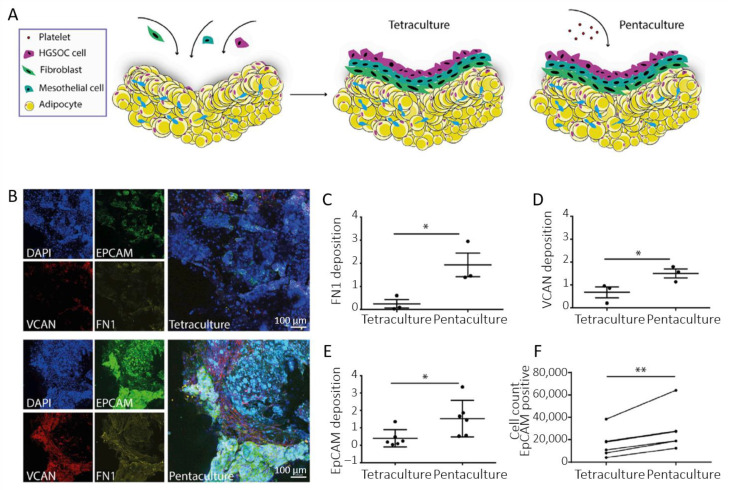
(**A**) Scheme of the multi-cellular setup. (**B**) Immunofluorescence staining of EpCAM-positive G164 cells, VCAN and FN1 in tetra- and penta-cultures. (**C**,**D**) Quantification of FN1 and VCAN deposition in the tetra- and penta-culture (at least 2 gels per condition, *n* = 3). (**E**) Quantification of EpCAM deposition. (**F**) Flow cytometry analysis of EpCAM-positive cells (*n* = 3). Data are expressed as mean ± SD (* *p* < 0.05 and ** *p* < 0.01; unpaired *t*-test for (**C**–**E**) and paired *t*-test for (**F**)). Adapted with permission from [105].

**Figure 7 biomolecules-13-00103-f007:**
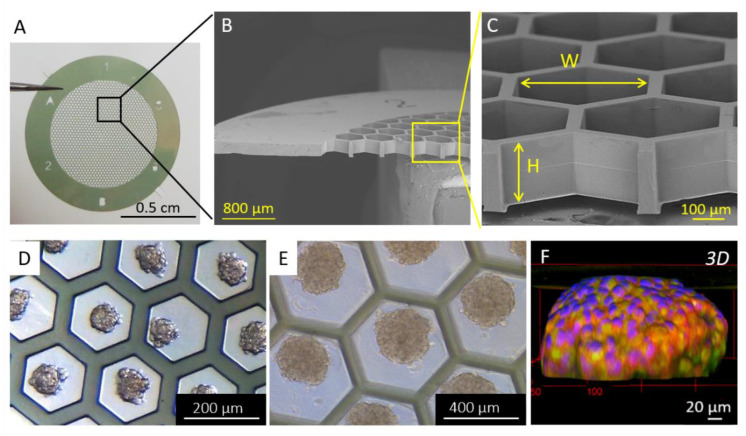
(**A**) Photograph and (**B**,**C**) SEM images of the bare patch. (**D**,**E**) Optical microscopy images of spheroids. (**F**) Immunofluorescence 3D reconstruction of a SKOV-3 spheroid with epithelial markers EpCAM (in green) and E-cadherin (in red), and nuclei (DAPI in blue). Reproduced with permission from [131].

**Figure 8 biomolecules-13-00103-f008:**
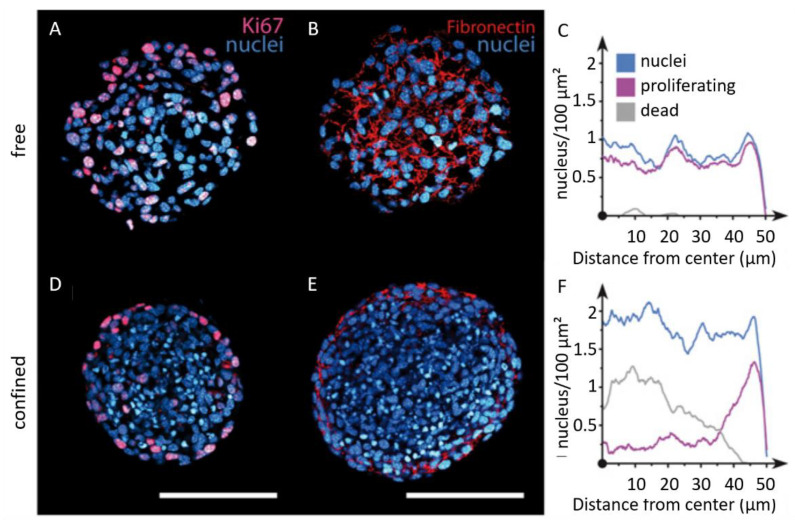
Imaging of the internal cellular organization of growing spheroids under elastic confinement. Confocal images of free (**A**,**B**) and confined (**D**,**E**) spheroids after cryosection and immunolabeling for DAPI (blue), KI67 (magenta), and fibronectin (red). Quantification of cell nuclei (blue), proliferating cells (purple), and dead cells (gray) radial densities for (**C**) free and (**F**) confined spheroids (scale bars 100 μm). Copyright (2013) National Academy of Sciences, USA [133].

**Figure 9 biomolecules-13-00103-f009:**
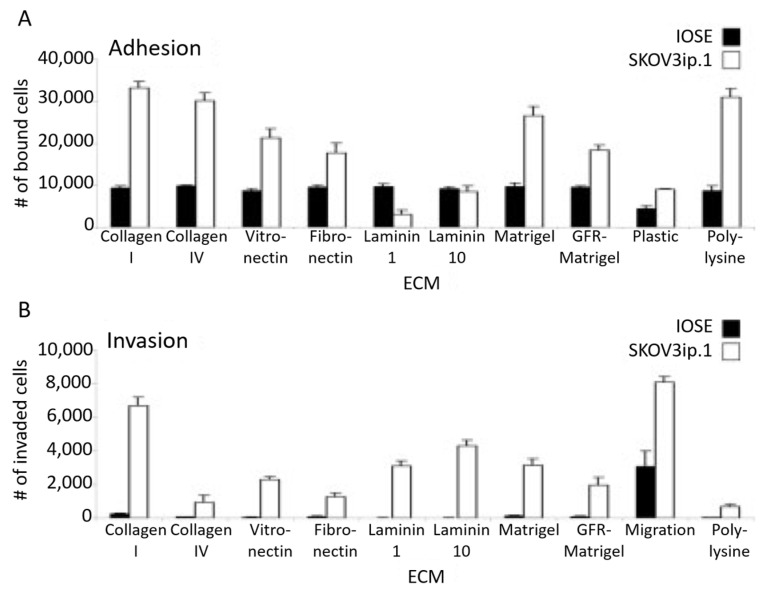
(**A**) Adhesion and (**B**) invasion assays of SKOV3ip.1 or IOSE fluorescently labeled cells on different ECM compositions. Reproduced with permission from [17].

**Figure 10 biomolecules-13-00103-f010:**
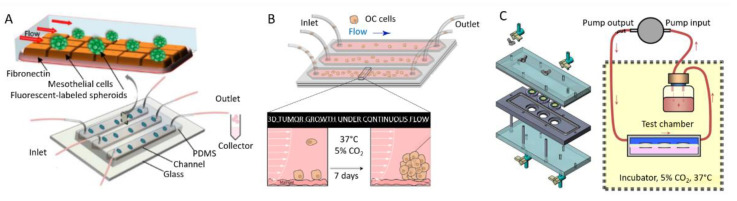
Schemes representing different microfluidic setups proposed in the literature. (**A**) 3D ovarian cancer–mesothelium microfluidic platform. Adapted with permission from [153]. (**B**) Matrigel-based linear microfluidic chip for investigating the impact of flow on 3D tumor growth. Reproduced with permission from the National Academy of Sciences, USA [154]. (**C**) Scheme of the experimental system and for application of wall shear stress on cultured cells. Reproduced with permission from [155].

**Figure 11 biomolecules-13-00103-f011:**
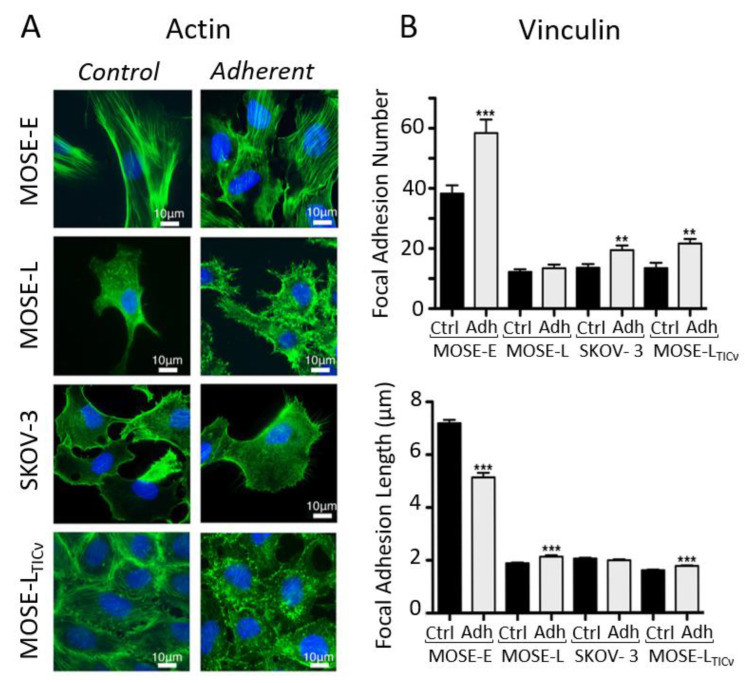
(**A**) Immunofluorescence images showing the reorganization of the actin (green) cytoskeleton in adherent MOSE-E, MOSE-L, SKOV-3, and MOSE-LTICv cells after exposure to fluid shear stress. (**B**) Quantitation of vinculin-positive focal adhesion number and size in control and after exposure to shear stress (*t*-test, ** *p* < 0.01, *** *p* < 0.001). Adapted with permission from [94].

**Figure 12 biomolecules-13-00103-f012:**
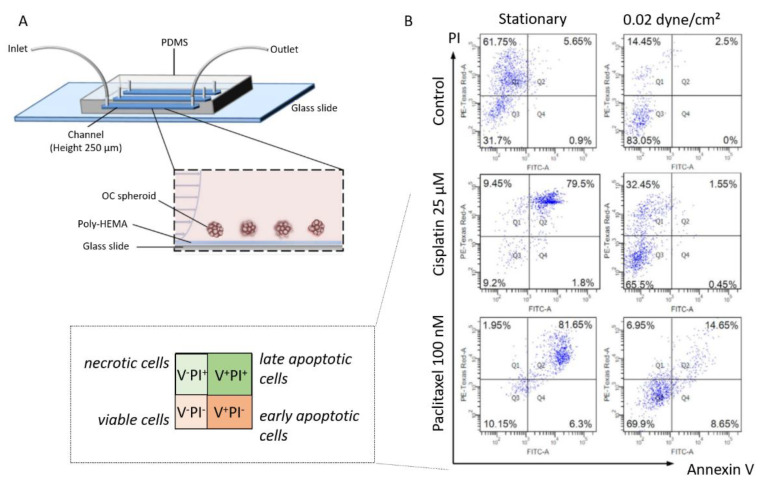
(**A**) Scheme of the experimental setup and side view of a poly-HEMA-coated (non-adherent) microfluidic channel under perfusion. (**B**) The chemosensitivity against cisplatin and paclitaxel of OC spheroids under static and 0.02 dyne/cm² shear stress was analyzed by Annexin V/PI staining. Adapted with permission from Springer Nature [96].

**Figure 13 biomolecules-13-00103-f013:**
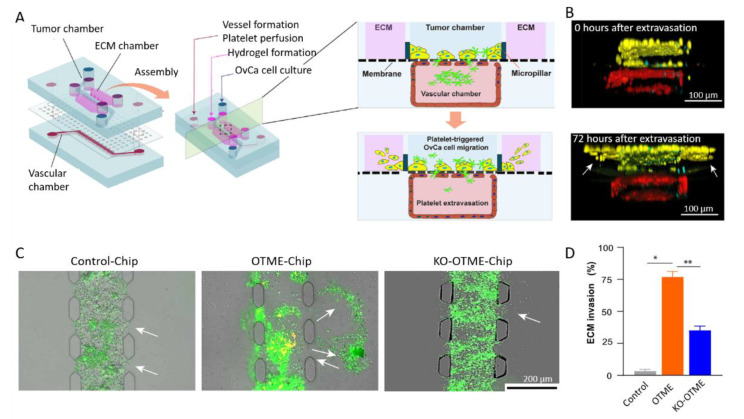
(**A**) Scheme of the microdevice containing two PDMS compartments separated by a thin, porous membrane mimicking the tumor–vascular interface. The right part shows a cross-sectional view of the chip. (**B**) Cross-section of 3D confocal imaging of OTME-Chip showing cancer cells (yellow), endothelial cells (red), and platelets (cyan) at 0 and 72 h after platelet extravasation. (**C**) Fluorescence microscopy images showing cancer cell (green) invasion (marked by arrows) into hydrogel ECM due to extravasated platelets (yellow). (**D**) Bar graph of the quantification of ECM invasion at 48 h (Dunnett test, * *p* < 0.05, ** *p* < 0.01). Adapted with permission from Science Advances [159].

**Figure 14 biomolecules-13-00103-f014:**
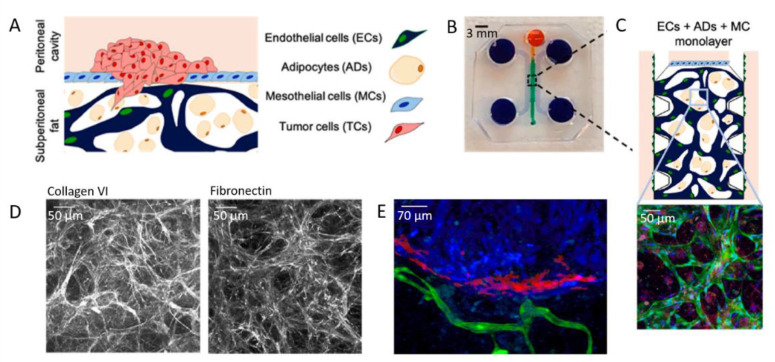
(**A**) Scheme of the 3D model and (**B**) PDMS mold with patterned channels fabricated using soft lithography. The central region (green) contains cells and a fibrin hydrogel. The side channels and reservoirs (purple) as well as the top channel and reservoir (orange) were filled with cell culture medium. (**C**) Scheme and confocal image of the vascularized model, in which ECs express GFP (green), nuclei are stained with DAPI (blue), and AD lipid droplets are stained with LipidTox (red). (**D**) Immunohistochemistry analysis of ECM. (**E**) 2D projected confocal z-stack of SKOV3 TCs (red) on the MC membrane (blue), but not invading the ECM of vascularized ECs (green) and ADs (not stained) on day 14, 7 days after high-density TC seeding. Adapted with permission from Elsevier [160].

## Data Availability

Not applicable.

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
