# Peer review of "In Vitro Models of Ovarian Cancer: Bridging the Gap between Pathophysiology and Mechanistic Models"

_biomolecules, 2023, doi:10.3390/biom13010103_

Round 1

Reviewer 1 Report

This review paper is quite clear and well-written. 

However, there are some fundamental and important aspects which are missing.

Despite the presence of 188 references, many of them are redundant while important and novel citations are missing.

Lately, a number of different papers on in-vitro models of ovarian cancer have been published, which are more complex and recapitulate better the human disease than the models described in this review (i.e. Ibrahim, Lina I et al. “Omentum-on-a-chip: A multicellular, vascularized microfluidic model of the human peritoneum for the study of ovarian cancer metastases.” Biomaterials vol. 288 (2022): 121728. doi:10.1016/j.biomaterials.2022.121728, Malacrida, Beatrice et al. “A human multi-cellular model shows how platelets drive production of diseased extracellular matrix and tissue invasion.” iScience vol. 24,6 102676. 29 May. 2021, doi:10.1016/j.isci.2021.102676 or Delaine-Smith, Robin M et al. “Modelling TGFβR and Hh pathway regulation of prognostic matrisome molecules in ovarian cancer.” iScience vol. 24,6 102674. 29 May. 2021, doi:10.1016/j.isci.2021.102674, and many more)

There are also other fundamental references missing, regarding CSC or the ECM (see Lupia, Michela et al. “Integrated molecular profiling of patient-derived ovarian cancer models identifies clinically relevant signatures and tumor vulnerabilities.” International journal of cancer vol. 151,2 (2022): 240-254. doi:10.1002/ijc.33983, Velletri, Tania et al. “Single cell-derived spheroids capture the self-renewing subpopulations of metastatic ovarian cancer.” Cell death and differentiation vol. 29,3 (2022): 614-626. doi:10.1038/s41418-021-00878-w, Pearce, Oliver M T et al. “Deconstruction of a Metastatic Tumor Microenvironment Reveals a Common Matrix Response in Human Cancers.” Cancer discovery vol. 8,3 (2018): 304-319. doi:10.1158/2159-8290.CD-17-0284 and many others)

In the description of the cellular environment of ovarian cancer, there is little, if none, mention about the role of the immune system and of the different components involved in the pathophysiology of this disease (i.e. the ascites fluid mentioned multiple times in this review is full of immune cells).

Overall, it is not clear whether the focus of the review is, primary or metastatic disease (and relative model) or both? If it is both all the points above mentioned need to be considered to provide a comprehensive review. 

Reviewer 2 Report

Review with great details of information and references.

I consider only a small review in English language. 

Please, follow my suggestion about the review:

In figure 1., I suggested the addition of reference or permition of use from the authors. 

In item  2.3.3., line 266 - Add more information about spheroids definition, organization, and their influence as ovarian cancer models.

Item 2.4.1. - Add which substrates or hidrogels were used in the studies presented. 

Item 3 - lines 412 and 420 - Add references. 

Best regards.

Reviewer 3 Report

This manuscript is an interesting review that attempts to shed light on new in vitro models to study ovarian cancer, a scientifically sound and relevant piece of field. The authors developed the work starting by describing the molecular characteristics of the tumor and outlining the development of highly innovative new models.  This is interesting in principle because of the need for new models in vitro to study this tumor at the mechanistic level and to test new therapeutic approaches, considering tumor/stroma. Overall, the study is well conducted in the second part of the manuscript, although I suggest adding results from new recent studies published in the last two years. However, I must outline some major concerns regarding the first part which appears as a list of the main characteristics of this tumor without any details and lacks critical interpretation of new and old data.

In this form, although interesting, this review offers a moderately important contribution to the field. I suggest shortening the first part of the manuscript and focalizing and improving on the second part, which is the title of the review.

Round 2

Reviewer 1 Report

This review has now been improved, but I will suggest to re-write some of the sentences (especially of the last paragraphs added in) as they are very very similar to some of the figure legends or part of the text present in the original papers. 

Reviewer 3 Report

The authors revised the manuscript accordingly with the requests that can be now accepted in the present form.

Author Response

We thank the reviewer for his/her comment and review on our revised manuscript.